

# Genetic characteristics of non-familial epilepsy

Kyung Wook Kang[1], Wonkuk Kim[2], Yong Won Cho[3], Sang Kun Lee[4], Ki-Young Jung[4], Wonchul Shin[5], Dong Wook Kim[6], Won-Joo Kim[7], Hyang Woon Lee[8], Woojun Kim[9], Keuntae Kim[3], So-Hyun Lee[10], Seok-Yong Choi[10] and Myeong-Kyu Kim[1]

[1] Department of Neurology, Chonnam National University Medical School, Gwangju, South Korea
[2] Department of Applied Statistics, Chung-Ang University, Seoul, South Korea
[3] Department of Neurology, Keimyung University Dongsan Medical Center, Daegu, South Korea
[4] Department of Neurology, Seoul National University Hospital, Seoul, South Korea
[5] Department of Neurology, Kyung Hee University Hospital at Gangdong, Seoul, South Korea
[6] Department of Neurology, Konkuk University School of Medicine, Seoul, South Korea
[7] Department of Neurology, Gangnam Severance Hospital, Yonsei University College of Medicine, Seoul, South Korea
[8] Department of Neurology, Ewha Womans University School of Medicine and Ewha Medical Research Institute, Seoul, South Korea
[9] Department of Neurology, Seoul St. Mary's Hospital, College of Medicine, The Catholic University of Korea, Seoul, South Korea
[10] Department of Biomedical Science, Chonnam National University Medical School, Gwangju, South Korea

Corresponding authors
Seok-Yong Choi, zebrafish@jnu.ac.kr
Myeong-Kyu Kim, mkkim@jnu.ac.kr

## ABSTRACT

**Background:** Knowledge of the genetic etiology of epilepsy can provide essential prognostic information and influence decisions regarding treatment and management, leading us into the era of precision medicine. However, the genetic basis underlying epileptogenesis or epilepsy pharmacoresistance is not well-understood, particularly in non-familial epilepsies with heterogeneous phenotypes that last until or start in adulthood.

**Methods:** We sought to determine the contribution of known epilepsy-associated genes (EAGs) to the causation of non-familial epilepsies with heterogeneous phenotypes and to the genetic basis underlying epilepsy pharmacoresistance. We performed a multi-center study for whole exome sequencing-based screening of 178 selected EAGs in 243 non-familial adult patients with primarily focal epilepsy (122 drug-resistant and 121 drug-responsive epilepsies). The pathogenicity of each variant was assessed through a customized stringent filtering process and classified according to the American College of Medical Genetics and Genomics guidelines.

**Results:** Possible causal genetic variants of epilepsy were uncovered in 13.2% of non-familial patients with primarily focal epilepsy. The diagnostic yield according to the seizure onset age was 25% (2/8) in the neonatal and infantile period, 11.1% (14/126) in childhood and 14.7% (16/109) in adulthood. The higher diagnostic yields were from ion channel-related genes and mTOR pathway-related genes, which does not significantly differ from the results of previous studies on familial or early-onset epilepsies. These potentially pathogenic variants, which were identified in genes that have been mainly associated with early-onset epilepsies with severe phenotypes, were also linked to epilepsies that start in or last until adulthood in this

study. This finding suggested the presence of one or more disease-modifying factors that regulate the onset time or severity of epileptogenesis. The target hypothesis of epilepsy pharmacoresistance was not verified in our study. Instead, neurodevelopment-associated epilepsy genes, such as *TSC2* or *RELN*, or structural brain lesions were more strongly associated with epilepsy pharmacoresistance.

**Conclusions:** We revealed a fraction of possible causal genetic variants of non-familial epilepsies in which genetic testing is usually overlooked. In this study, we highlight the importance of earlier identification of the genetic etiology of non-familial epilepsies, which leads us to the best treatment options in terms of precision medicine and to future neurobiological research for novel drug development. This should be considered a justification for physicians determining the hidden genetics of non-familial epilepsies that last until or start in adulthood.

## INTRODUCTION

Epilepsy is one of the most common neurological conditions affecting approximately eight of every 1,000 individuals worldwide (*Fiest et al., 2017*). Although its detailed pathogenesis remains largely unknown, a cumulative understanding of the genetic basis of epilepsy revealed that many epilepsies that were previously considered idiopathic should be reclassified as having a genetic cause (*Thomas & Berkovic, 2014*). Even acquired epilepsies resulting from trauma, stroke, neoplasm, infection, or congenital malformation are now known to be associated with genetic contributions (*Thomas & Berkovic, 2014*). Indeed, hundreds of genes have already been associated with epilepsy to date (*Wang et al., 2017*), and have now been incorporated into commercial clinical tests with comprehensive gene panels for the rapid identification of causative genetic mutations of many forms of epilepsy (*Møller et al., 2016*; *Hildebrand et al., 2016*; *Dunn et al., 2018*). This is highly important, because knowledge of the genetic etiology can provide essential prognostic information and influence decisions regarding treatment and management, leading us into the era of precision medicine (*Milligan et al., 2014*; *Pierson et al., 2014*; *Lindy et al., 2018*).

Unlike neonatal- and childhood-onset epilepsy, in which both availability of genetic testing and the actionability of test results are higher (*Møller et al., 2016*), enquiry into genetic causes of epilepsy has been overlooked in adult patients with epilepsy (APEs) for a number of reasons (*Thomas & Berkovic, 2014*): underappreciation of the role of genetic factors in certain epilepsies such as adult-onset focal epilepsy, an inexact causal attribution such as mistakenly ascribing a developmental epileptic encephalopathy (DEE) to birth trauma and, not least, unknown family history resulting from the absence of the oldest living relative who tends to be the most accurate custodian of family history or excessive social stigma leading to non-disclosure of seizures in the patient's older relatives. It is also notable that most non-familial APEs in practice are not willing to submit their unaffected family members to genetic testing, resulting in the inheritance pattern of the family often being inconclusive. Furthermore, in most APEs, particularly those who are

not candidates for presurgical evaluations, either voluntarily or involuntarily, the detailed epilepsy phenotypes are generally indistinct. All of these factors have contributed to reluctance in genetic testing of APEs in practice, delaying our understanding of the genetic basis of non-familial epilepsies and preventing APEs from having the opportunity to receive potentially better treatment options.

However, the paradigm of genetically diagnosing non-familial APEs has shifted with advances in sequencing technology. It is now well-known that genetic diagnosis is no longer an exclusive property of certain familial Mendelian epilepsies. For example, post-zygotic de novo mutations have been discovered in some sporadic focal epilepsies or DEEs, thus indicating genetic causation in patients with epilepsy even without a family history (*Phillips et al., 2000*; *Claes et al., 2001*; *Bisulli et al., 2004*; *Nava et al., 2014*). Furthermore, this paradigm shift provides us with optimistic but reasonable prospects. There might be undetermined causal variants in non-familial APEs, particularly those experiencing earlier onset of seizures, as their epilepsy diagnosis was likely made in the non-genomic era when adequate genetic testing was not available, and as such were not genetically diagnosed. In addition, there might be a hidden native genetic basis of non-familial adult-onset epilepsy, as suggested by surprising genetic causes in pediatric patients with non-familial DEEs (*Claes et al., 2001*; *Nava et al., 2014*).

The higher diagnostic yield of genetic testing in DEEs has been associated with primarily drug-refractory seizures (*Møller et al., 2016*; *Ko et al., 2018*; *Rim et al., 2018*), which indicates that causal genes of DEEs could be linked to pharmacoresistance. Indeed, the target hypothesis is one of the most frequently cited theories of epilepsy pharmacoresistance, and postulates that alterations in the properties of antiepileptic drug (AED) targets, such as compositional changes in voltage-gated ion channels and neurotransmitter receptors, result in decreased drug sensitivity and thus leads to refractoriness (*Tang, Hartz & Bauer, 2017*). Interestingly, the genes encoding the voltage-gated ion channels and neurotransmitter receptors have also been most commonly associated with epilepsy (*Wang et al., 2017*; *Lindy et al., 2018*). This indicates that there might be a common pathway underlying both epileptogenesis and epilepsy pharmacoresistance.

In the present study, we sought to determine the contribution of known epilepsy-associated genes (EAGs) to the causation of non-familial epilepsies with heterogeneous phenotypes and to the genetic basis underlying epilepsy pharmacoresistance.

## MATERIALS AND METHODS

### Study design and participants

In this multi-center study, consecutive patients with an established clinical diagnosis of epilepsy as defined by a practical clinical definition of epilepsy (*Fisher et al., 2014*) and who had been managed by epilepsy specialists over a period of 2 years were recruited from 10 tertiary epilepsy referral centers in Korea. All study participants were eligible if they had drug-resistant (DR group) or drug-responsive (DS group) epilepsy according to the following definitions and criteria. To enhance the contrast of phenotype between DS and

DR group, we defined drug resistance more stringently than the conventional definition (*Kwan et al., 2010*) as the occurrence of at least 12 unprovoked seizures over the course of 1 year before recruitment, with trials of two or more appropriate AEDs at the maximal tolerated doses, which were established on the basis of the occurrence of clinical side effects at supramaximal doses. Patients who underwent surgical treatment for DR group epilepsy were classified as having DR group epilepsy, regardless of the surgical outcome. In patients treated with a single AED, drug responsiveness was defined as complete freedom from seizures for at least 1 year up to the date of the last follow-up visit. However, patients who had a definite history of epilepsy in first- or second-degree relatives, were frequently in poor compliance with AED therapy, had reported non-motor seizures only without consciousness impairment, or had progressive DEEs were excluded.

An extensive historical assessment was performed in all participants using a standardized form, detailing the epidemiology, seizure characteristics, epilepsy syndrome, electroencephalography and magnetic resonance imaging findings, family history, treatment, and treatment-emergent adverse events.

This study was approved by the institutional review boards at Chonnam National University Hospital (CNUH-20160028). All research was performed in accordance with relevant guidelines and regulations, and written informed consent was obtained from all study participants.

## Whole exome sequencing

Following genomic DNA (gDNA) extraction from whole blood, the Agilent SureSelect Target Enrichment protocol for Illumina paired-end sequencing (ver. B.3, June 2015; Agilent Technologies, Santa Clara, CA, USA) was used together with 200 ng input gDNA for the generation of standard exome capture libraries. In all cases, the SureSelect Human All Exon V5 probe set was used. For exome capture, 250 ng of DNA library was mixed with hybridization buffers, blocking mixes, RNase inhibitors, and five μl of the SureSelect all exon capture library, according to the standard Agilent SureSelect Target Enrichment protocol. Hybridization to the capture baits was conducted at 65 °C using the heated thermal cycler lid option at 105 °C for 24 h on a polymerase chain reaction (PCR) machine. The captured DNA was amplified, purified, quantified and then sequenced using the HiSeq™ 2,500 platform (Illumina, San Diego, CA, USA). For sequence alignment, paired-end sequences were first mapped to the human genome (UCSC assembly hg19; original GRCh37 from NCBI, February 2009) using BWA (Burrows-Wheeler Alignment Tool, v0.7.12). The programs packaged in PicardTools (v1.130; Broad Institute, Cambridge, MA, USA) were then applied to remove PCR duplicates. Base quality score recalibration and local realignment around indels were performed using the Genome Analysis Toolkit (GATK; Broad Institute, Cambridge, MA, USA) to locally realign reads such that the number of mismatching bases was minimized across all reads. Based on the previously generated binary alignment map file, variant genotyping for each sample was performed using the Haplotype Caller in the GATK. Those variants are annotated by another program called SnpEff (v4.1g, http://snpeff.sourceforge.net/), converted to the vcf file format, filtered through the single nucleotide polymorphism (SNP) database

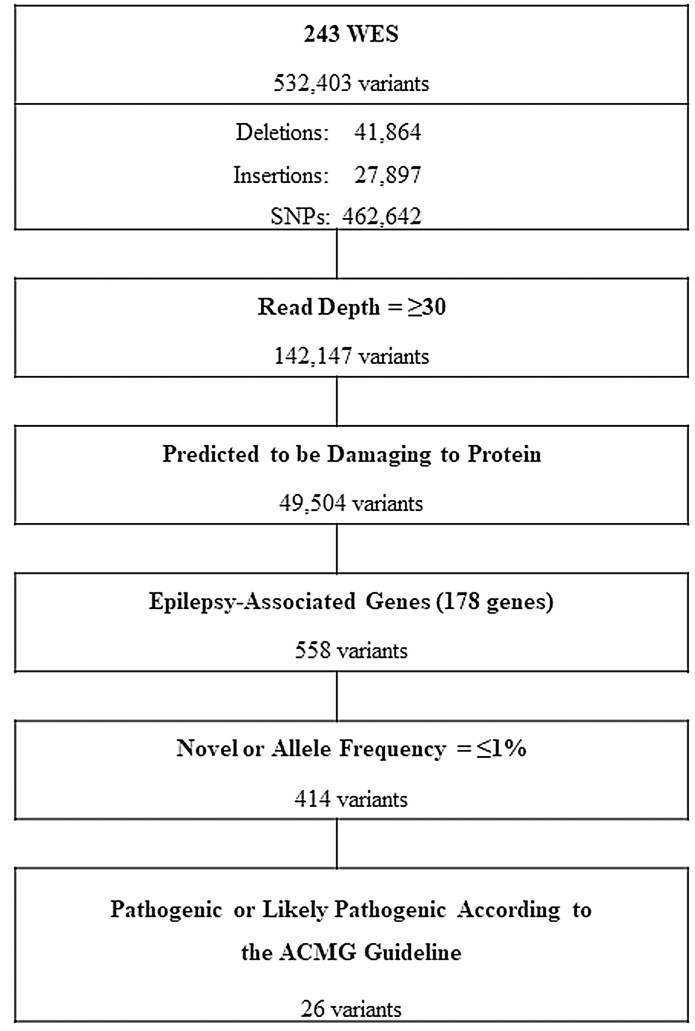

**Figure 1 Workflow of variants filtering process.** WES, whole exome sequencing; SNPs, single nucleotide polymorphisms; ACMG, American College of Medical Genetics and Genomics.

(dbSNP, v142) and compared to SNPs from the 1,000 Genome Projects. Our in-house program and SnpEff were then applied to filter the data through additional databases, including ESP6500, ClinVar and dbNSFP2.9.

## Whole exome sequencing interpretation

The workflow for whole exome sequencing (WES) data interpretation to identify high confidence candidate variants with higher predicted potential for pathogenicity in epilepsy is provided in Fig. 1. Briefly, variants satisfying all of the following conditions were further analyzed: variants with a read depth of ≥30×, variants predicted to be disruptive or damaging to the protein for which they code (frame-shifted, nonsense, non-synonymous missense, small indels, or canonical splice site variants) and variants of 178 known EAGs (Table 1). The selection criteria of the EAGs were as follows: (1) epilepsy genes that cause pure or relatively pure epilepsies or syndromes with epilepsy as the core symptom
**Table 1 Epilepsy associated genes.**

**Epilepsy genes** *AARS, ADRA2B, ADSL, ALDH7A1 ALG13, ARV1, ATP6AP2, CACNA1A, CACNA1H, CACNB4, CASR, CDKL5, CERS1, CHD2, CHRNA2, CHRNA4, CHRNB2, CLCN2, CLN3, CLN5, CLN6, CLN8, CNTN2, CPA6, CSTB, CTSD, DEPDC5, DNM1, DOCK7, EEF1A2, EFHC1, EPM2A, FGF12, FOXG1, FRRS1L, GABRA1, GABRB1, GABRB3, GABRD, GABRG2, GAL, GAMT, GATM, GNAO1, GOSR2, GPR98, GRIN2A, GRIN2B, GRIN2D, GUF1, HCN1, ITPA, KCNA2, KCNB1, KCNC1, KCNMA1, KCNQ2, KCNQ3, KCNT1, KCTD7, LGI1, LMNB2, MFSD8, NECAP1, NHLRC1, NPRL2, NPRL3, NRXN1, PCDH19, PLCB1, PNPO, POLG, PPT1, PRDM8, PRICKLE1, PRIMA1, PRRT2, SCARB2, SCN1A, SCN1B, SCN2A, SCN8A, SCN9A, SIK1, SLC12A5, SLC13A5, SLC1A2, SLC25A12, SLC25A22, SLC2A1, SLC6A1, SLC9A6, SPTAN1, ST3GAL3, ST3GAL5, STX1B, STXBP1, SZT2, TBC1D24, TCF4, TPP1, UBA5, UBE3A, WWOX, ZEB2*

**Neurodevelopment-associated epilepsy genes** *ANKLE2, AMPD2, ARFGEF2, ARX, ASPM, ATN1, CASK, CCDC88C, CDK5, CENPE, CENPJ, CLP1, CNTNAP2, COL4A2, DCX, DIAPH1, EMX2, ERMARD, EXOSC3, FIG4, FLNA, GPR56, HERC1, IER3IP1, KATNB1, KIF11, KIF2A, KIF5C, LAMB1, LAMC3, MED17, MFSD2A, MPDZ, NDE1, NSDHL, OCLN, OPHN1, PAFAH1B1, PCLO, PIK3R2, PLEKHG2, PNKP, PPP1R15B, PTCH1, QARS, RELN, RTTN, SASS6, EPSECS, SLC12A6, SLC20A2, SNIP1, SPATA5, SRPX2, STAMBP, STRADA, SYN1, TRMT10A, TSC1, TSC2, TSEN15, TSEN2, TSEN54, TUBA1A, TUBA8, TUBB2A, TUBB2B, TUBB3, TUBG1, VPS53, WDR62, WDR73, XPR1*

($n$ = 105) and (2) neurodevelopment-associated epilepsy genes that produce gross neurodevelopmental malformation and epilepsy ($n$ = 73), which may vary in severity (*Perucca et al., 2017*; *Wang et al., 2017*).

Of the selected variants, variants with a minor-allele frequency of >1% in the Korean Reference Genome Database (KRGDB; http://152.99.75.168/KRGDB/) or Exome Aggregation Consortium (ExAC; http://exac.broadinstitute.org/) were excluded from further analysis, as an allele frequency in a control population that is, greater than expected for the disorder is considered strong support for a benign interpretation (*Richards et al., 2015*).

The deleteriousness of the selected variants was predicted by 11 current deleteriousness-scoring methods, including eight function prediction methods (Polyphen-2_HDIV, Polyphen-2_HVAR, SIFT, MutationTaster, Mutation Assessor, LRT, FATHMN and PROVEAN), one conservation score method (GERP++) and two ensemble score methods (MetaSVM and MetaLR). The variants predicted by two or more prediction scores as deleterious or damaging to the protein for which they code were included in further analysis. The pairs of prediction scores, Polyphen-2_HDIV and Polyphen-2_HVAR and MetaSVM and MetaLR, received a single score each in the scoring of the deleteriousness of a variant because the two prediction scores in each pair have a strong linear correlation (*Dong et al., 2015*; *Liu et al., 2016*). Known pathogenic variants or synonymous variants causing the same amino acid change were determined by searching ClinVar (https://www.ncbi.nlm.nih.gov/clinvar/) and the latest professional version of the Human Gene Mutation Database (http://www.hgmd.cf.ac.uk/). Any inconsistency among the sources was considered as uncertain in the functional significance of the variants.

The final variants selected via the filtering steps were classified using a five-class scheme of pathogenicity (pathogenic, likely pathogenic, uncertain significance, benign, or likely benign) according to the latest guidelines for the interpretation of sequence variants by the American College of Medical Genetics and Genomics (ACMG) (*Richards et al., 2015*). Among the variants classified as pathogenic or likely pathogenic (P/LPs), a heterozygous variant alone in exclusively recessive genes presenting as a typical recessive disorder was tested for compound heterozygosity using CNVkit (*Talevich et al., 2016*) for copy number detection. All variants selected as P/LPs were validated by Sanger sequencing.

**Table 2 Characteristics of the study participants.**

|  | Drug-responsive (n = 121) | Drug-resistant (n = 122) | p-value |
|---|---|---|---|
| **Age (years)** | | | |
| At recruitment | 39.3 ± 15.1 (range, 20–84) | 39.9 ± 11.3 (range, 20–68) | 0.706 |
| At seizure onset | 25.4 ± 15.2 (range, 0–68) | 15.9 ± 10.1 (range, 0–45) | <0.001 |
| Gender (male) | 65 (53.7%) | 64 (52.5%) | 0.898 |
| **Epilepsy classification** | | | |
| Generalized | 19 (15.7%) | 4 (3.3%) | |
| Focal | 95 (78.5%) | 118 (96.7%) | <0.001 |
| Unknown | 7 (5.8%) | 0 (0%) | |

## Statistics

Of the approximately 600 APEs that were consecutively enrolled in this study, age- and gender-matched APEs were randomly assigned to the DR and DS groups. The two groups were compared by Fisher's exact test for categorical data or Student's t-test for continuous data. A p-value of <0.05 was considered significant. The Statistical Package for the Social Sciences (v23.0; SPSS, Chicago, IL, USA) was used for all analyses.

To determine whether the prevalence of the selected EAG variants in APEs was significantly increased compared to the control group, the odds ratio (OR) of each variant was calculated using the adjusted allele frequency of the variant in the ExAC database. If the OR was >5.0 and the confidence interval (CI) around the estimate of the OR did not include 1.0, the difference in prevalence between the groups was considered to be statistically significant (*Richards et al., 2015*).

## RESULTS

### Participant characteristics

A total of 243 APEs (121 in the DS group and 122 in the DR group) were randomized and their epidemiological and clinical characteristics are provided in Table 2. Briefly, the mean ages at recruitment and at seizure onset were approximately 40 (median; 38, range; 20–84) and 20 (median; 17, range; 0–68) years, respectively. According to the seizure onset age, 3.3% (8/243) experienced their first seizure in the neonatal and infantile period (aged 0–1 year), 51.9% (126/243) in childhood (aged 2–18 years) and 44.9% (109/243) in adulthood (aged >19 years). The mean age at seizure onset was significantly different between the DS and DR groups, but was similar between APEs with and without P/LPs (21.1 ± 14.4 and 20.6 ± 13.7 years, respectively).

### Identification of pathogenic variants

All participants underwent high-coverage WES and yielded a total of 532,403 variants from which, after a customized stringent six-step filtering process (Fig. 1), 26 variants in 15 EAGs were determined to be P/LPs (three pathogenic and 23 likely pathogenic) according to the ACMG guideline (*Richards et al., 2015*) in 32 of 243 APEs (13.2%) (Table 3).

**Table 3 Pathogenic or likely pathogenic variants according to the ACMG guideline.**

| Gene | Chr. | Position | HGVS.p | OR | 95% CI | ACMG Criteria | Interpretation |
|------|------|----------|--------|-----|--------|---------------|----------------|
| ADGRV1 | chr5 | 89,977,183 | p.His1859Arg | 11.7 | 3.6–37.6 | PS4, PM6, PP3, PP5 | Likely pathogenic |
| CHRNA4 | chr20 | 61,981,730 | p.Arg345Cys | NA | NA | PS1, PS3, PM2, PM6, PP3 | Pathogenic |
| CNTNAP2 | chr7 | 146,741,111 | p.Ile172Thr | 35.7 | 4.3–290.5 | PS4, PM6, PP3, PP5 | Likely pathogenic |
| | chr7 | 148,112,574 | p.Arg1288Cys | 83.3 | 8.6–802.2 | PS4, PM6, PP3, PP5 | Likely pathogenic |
| DEPDC5 | chr22 | 32,215,100 | p.Arg587X | NA | NA | PVS1, PS3, PM2, PM6, PP3 | Pathogenic |
| | chr22 | 32,242,890 | p.Pro1031His | 7.0 | 1.7–28.7 | PS4, PM6, PP3 | Likely pathogenic |
| EFHC1 | chr6 | 52,319,049 | p.Arg294Cys | 125.2 | 11.3–1382.9 | PS4, PM5, PP3, PP5 | Likely pathogenic |
| GABRG2 | chr5 | 161,495,029 | p.Ser8Arg | 250.6 | 35.2–1782.6 | PS4. PM6, PP3, PP5 | Likely pathogenic |
| HCN1 | chr5 | 45,695,898 | p.Ser100Ala | 235.5 | 14.7–3770.7 | PS4, PM5, PM6, PP3 | Likely pathogenic |
| KCNB1 | chr20 | 47,990,709 | p.Ile463Thr | 14.7 | 1.9–110.8 | PS4, PM6, PP3 | Likely pathogenic |
| KCNT1 | chr9 | 138,670,613 | p.Glu892Lys | 24.9 | 3.1–194.6 | PS4, PM6, PP3, PP5 | Likely pathogenic |
| PRICKLE1 | chr12 | 42,858,215 | p.Ala541Ser | 376.9 | 62.8–2260.6 | PS4, PM6, PP3, PP5 | Likely pathogenic |
| RELN | chr7 | 103,197,510 | p.Thr1904Met | 23.6 | 5.5–100.9 | PS4, PM6, PP3, PP5 | Likely pathogenic |
| | chr7 | 103,276,733 | p.Lys751Thr | 9.6 | 1.3–71.1 | PS4, PM6, PP3, PP5 | Likely pathogenic |
| SCN1A | chr2 | 166,850,785 | p.Arg1575Cys | 55.3 | 11.9–256.8 | PS4, PM6, PP3, PP5 | Likely pathogenic |
| | chr2 | 166,903,464 | p.Thr398Met | 250.3 | 15.6–4007.8 | PS4, PM6, PP3 | Likely pathogenic |
| | chr2 | 166,894,321 | p.Val971Ile | 55.3 | 11.9–256.7 | PS4, PM6, PP3, PP5 | Likely pathogenic |
| SCN9A | chr2 | 167,141,015 | p.Asn641Ser | 123.3 | 11.2–1362.3 | PS4, PM5, PM6, PP3 | Likely pathogenic |
| TSC1 | chr9 | 135,771,689 | p.Pro1143Leu | 83.4 | 8.7–803.3 | PS4, PM6, PP3, PP5 | Likely pathogenic |
| | chr9 | 135,772,927 | p.Thr899Ser | 41.8 | 9.3–187.3 | PS4, PM6, PP3, PP5 | Likely pathogenic |
| | chr9 | 135,776,993 | p.Ser829Arg | 62.4 | 13.2–294.6 | PS4, PM6, PP3, PP5 | Likely pathogenic |
| TSC2 | chr16 | 2,134,649 | p.Glu1476Gln | 62.1 | 6.9–556.7 | PS4, PM5, PM6, PP3, PP5 | Pathogenic |
| | chr16 | 2,135,247 | p.Arg1529Gln | 13.3 | 1.8–100.2 | PS4, PM6, PP3, PP5 | Likely pathogenic |
| | chr16 | 2,127,648 | p.Val963Met | 41.7 | 5.0–347.2 | PS4, PM6, PP3, PP5 | Likely pathogenic |
| | chr16 | 2,129,146 | p.Leu1027Pro | NA | NA | PM2, PM6, PP3, PP5 | Likely pathogenic |
| | chr16 | 2,134,692 | p.Glu1490Gly | 14.5 | 1.9–108.7 | PS4, PM6, PP3, PP5 | Likely pathogenic |

**Note:**
ACMG, American College of Medical Genetics and Genomics; Chr, chromosome; HGCV.p, Human Genome Variation Society nomenclature for protein; OR, odds ratio; CI, confidence interval; NA, not available.

The diagnostic yield according to seizure onset age was 25% (2/8) in the neonatal and infantile period, 11.1% (14/126) in childhood and 14.7% (16/109) in adulthood (Table 4).

Three of the twenty-six P/LPs identified in this study were novel variants (absent from controls in the ExAC database), and the remaining 23 P/LPs were known but extremely rare variants of which the mean OR was 85.5 (range; 7.02–376.9) and the CI around each estimate of the OR did not include one. The classification criteria for the pathogenicity of each P/LP applied according to the ACMG guideline in this study are described in Table 3. Thirty heterozygous variants classified as P/LPs of 19 recessive genes (ALDHDA1, ASPM, CCDC88C, CENPJ, CLN3, CLN8, GPR56, LAMB1, MECP2, MFSD8, NHLRC1, NRXN1, POLG, RTTN, SLC12A6, TBC1D24, TRMT10A, TUBA8 and WWOX) were not included in the diagnostic yield calculation.

**Table 4 Presumed disease-causative genes of non-familial epilepsies.**

| P/LP variants[†] | | Pt_ID[‡] | Sex/Age[*], years | Drug response | Febrile seizure | Epilepsy classification | Etiology |
|---|---|---|---|---|---|---|---|
| ION CHANNEL-RELATED GENES | | | | | | | |
| CHRNA4 | **p.Arg345Cys** | DK085 | M/26 (12) | DS | N | Focal | Non-lesional |
| GABRG2 | p.Ser8Arg | JN086 | M/22 (4) | DR | N | Focal | Tumor |
| | | JN167 | M/68 (10) | DR | Y | Focal | Non-lesional |
| HCN1 | p.Ser100Ala | SC009 | M/24 (15) | DS | N | Focal | Non-lesional |
| KCNB1 | p.Ile463Thr | JN134 | F/52 (29) | DS | Y | Focal | FCD |
| KCNT1 | p.Glu892Lys | **SU059** | M/25 (20) | DR | N | Focal | Non-lesional |
| SCN1A | p.Thr398Met | JN129 | F/43 (29) | DR | N | Focal | HS |
| | p.Val971Ile | JN168 | M/30 (1) | DR | N | Focal | Non-lesional |
| | | KG012 | M/43 (38) | DR | N | Focal | Trauma |
| | p.Arg1575Cys | JN046 | F/54 (16) | DS | N | Focal | Non-lesional |
| | | JN166 | F/43 (29) | DS | N | Focal | Non-lesional |
| SCN9A | p.Asn641Ser | DK098 | F/35 (12) | DR | NA | Focal | HS |
| mTOR PATHWAY-RELATED GENES | | | | | | | |
| DEPDC5 | **p.Arg587X** | DK023 | F/26 (19) | DS | N | Focal | Non-lesional |
| | p.Pro1031His | **SU059** | M/25 (20) | DR | N | Focal | Non-lesional |
| | | JN114 | M/38 (11) | DS | N | Focal | Non-lesional |
| TSC1 | p.Ser829Arg | SU036 | M/40 (1) | DR | N | Focal | FCD |
| | | KH015 | F/45 (37) | DS | N | Focal | HS |
| | p.Thr899Ser | **SU023** | M/33 (21) | DR | N | Focal | FCD |
| | | **JN036** | M/51 (41) | DS | Y | Focal | Trauma |
| | p.Pro1143Leu | **JN224** | F/65 (55) | DS | N | Focal | Encephalitis |
| TSC2 | p.Val963Met | KH016 | F/42 (34) | DR | N | Focal | HS |
| | p.Leu1027Pro | **JN056** | M/37 (7) | DR | N | Focal | TS |
| | **p.Glu1476Gln** | JN051 | M/49 (28) | DR | N | Focal | HS |
| | p.Glu1490Gly | EW001 | F/64 (8) | DR | N | Focal | HS |
| | p.Arg1529Gln | JN006 | F/31 (18) | DS | N | Focal | Non-lesional |
| ADHESION MOLECULE/RECEPTOR-RELATED GENES | | | | | | | |
| ADGRV1 | p.His1859Arg | **JN036** | M/51 (41) | DS | Y | Focal | Trauma |
| | | JN023 | F/25 (5) | DR | Y | Focal | HS |
| | | DK066 | F56 (46) | DS | N | Generalized | Non-lesional |
| CNTNAP2 | p.Ile172Thr | **SU023** | M/33 (21) | DR | N | Focal | FCD |
| | p.Arg1288Cys | JN041 | M/38 (17) | DR | N | Focal | Non-lesional |
| SIGNAL TRANSDUCTION-RELATED GENES | | | | | | | |
| EFHC1 | p.Arg294Cys | JN172 | M/36 (32) | DS | Y | Focal | Trauma |
| PRICKLE1 | p.Ala541Ser | JN072 | M/60 (33) | DR | Y | Focal | Non-lesional |
| | | **JN224** | F/65 (55) | DS | N | Focal | Encephalitis |
| | | YC009 | M/34 (2) | DR | Y | Focal | Non-lesional |
| EXTRACELLULAR MATRIX-RELATED GENES | | | | | | | |
| RELN | p.Lys751Thr | SU018 | F/44 (7) | DR | N | Focal | Non-lesional |
| | p.Thr1904Met | DK021 | F/44 (25) | DR | N | Focal | HS |
| | | **JN056** | M/37 (7) | DR | N | Focal | TS |

**Notes:**
[†] Bold denotes variants classified as pathogenic.
[‡] Bold denotes participant with two P/LPs.
[*] Age at recruitment (at seizure onset).
Abbreviations: DR, drug refractory group; DS, drug responsive group; FCD, focal cortical dysplasia; HS, hippocampal sclerosis; NA, not available; P/LP, pathogenic/likely pathogenic variant; TS, tuberous sclerosis; mTOR, mammalian target of rapamycin.

## Presumed disease-causative genes of non-familial epilepsies

Three variants were classified as pathogenic, including *CHRNA4* p.Arg345Cys and two variants of mTOR pathway-related genes (*DEPDC5* p.Arg586X and *TSC2* p.Glu1476Gln). Among the 23 variants classified as likely pathogenic, eight were variants of ion channel-related genes (*GABRG2, HCN1, KCNB1, KCNT1, SCNIA* and *SCN9A*), eight of mTOR genes (*DEPDC5, TSC* and *TSC2*), three of cell adhesion molecule/receptor-related genes (*ADGRV1* and *CNTNAP2*), two of extracellular matrix-related genes (*RELN*) and two of signal transduction-related genes (*EFHC1* and *PRICKLE1*) (Table 4) (*Myers & Mefford, 2015*; *Wang et al., 2017*; *GeneCards, 2018*). Three genes (*SCNIA, TSC1* and *TSC2*) were found to have a higher diagnostic yield of genetic testing, with each accounting for 15.6% (5/32) of the total yield. Five of two hundred and forty-three APEs (2.1%) had two independent P/LPs simultaneously, the functional categories of which differed from each other (Table 4). All five APEs with two P/LPs simultaneously had one of the mTOR gene variants.

## Pathogenic potential of EAGs in AED responsiveness

The diagnostic yield was 10.7% in the DS group and 15.6% in the DR group ($p > 0.05$). Six genes were commonly associated with both the DS and the DR group, including *ADGRV1, DEPDC5, PRICKLE1, SCNIA, TSC1* and *TSC2*. Structural brain lesions were seen in 17 of the 32 APEs (53.1%) with P/LPs, which are highly likely to have caused their epilepsies, whereas 63.2% of the DR group but 38.5% of the DS group had potentially causal lesions.

Two APEs with *SCN1A* p.Arg1575Cys were seizure-free with monotherapy with carbamazepine (CBZ) while three APEs with *SCN1A* p.Thr398Met or p.Val971Ile were resistant to drugs with various mechanisms of action, including CBZ. Four of five P/LPs of *TSC2* were associated with DR group epilepsy, while drug responsiveness differed even among patients with the same variant of *TSC1*. All three APEs with *RELN* variants were multi-drug resistant (Table 4).

## Genotype-phenotype correlation

Thirty-one of the thirty-two APEs with P/LPs had focal epilepsies. Of the three APEs with the *ADGRV1* p.His1859Arg variant, two were diagnosed with focal epilepsy and one with generalized epilepsy.

In five of the 12 APEs with ion channel-related gene variants, potentially disease-causative lesions were identified, including a tumor, focal cortical dysplasia (FCD), hippocampal scleroses (HS) and traumatic brain tissue loss. Two had a definite history of febrile seizures (FS) and one (JN168 in Table 4) had a history of what was considered to be an early infantile EE (i.e., seizure onset during infancy, autistic behaviors, mental retardation and multi-drug resistant seizures).

Eight of ten APEs (80%) with *TSC1* or *TSC2* variants but none of the three APEs with *DEPDC5* variants had brain malformations including FCD, HS, or TS. Only one of ten APEs with *TSC1* or *TSC2* variants had clinical presentations fitting the diagnostic criteria of tuberous sclerosis complex (*Samueli et al., 2015*). In six of the eight APEs with

malformations, the seizures were multi-drug resistant and the mean duration from seizure onset to genetic diagnosis was approximately 27.7 years. One of the thirteen APEs with P/LPs of mTOR genes had a history of FS (Table 4).

## DISCUSSION

### Possible causal genetic variants of non-familial epilepsy

In the present study, we discovered possible causal genetic variants in 13.2% (32/243) of non-familial epilepsy cases. Insofar as non-familial focal epilepsy only and non-familial adulthood-onset epilepsy only were concerned, the diagnostic yields were 14.6% (31/213) and 14.7% (16/109), respectively. Although other study designs varied such that direct comparison to our study may not be suitable, there was a distinct tendency of higher genetic yields to associate with early childhood epilepsy with a distinct phenotype such as early onset DEEs, a positive history of familial epilepsy, or a generalized epilepsy (*Lemke et al., 2012*; *Carvill et al., 2013*; *Kodera et al., 2013*; *Wang et al., 2014*; *Della Mina et al., 2015*; *Mercimek-Mahmutoglu et al., 2015*; *Hildebrand et al., 2016*; *Møller et al., 2016*; *Dunn et al., 2018*; *Ko et al., 2018*; *Rim et al., 2018*; *Lee, Lee & Lee, 2018*). Given that the present study examined primarily non-familial focal epilepsies with heterogeneous phenotypes, of which almost half were adulthood-onset epilepsies, and adopted WES for genetic testing that encompasses only a proportion of all mutations, the genetic yields found in our study were beyond our expectation. This should be considered a justification for physicians determining potentially causal genetic variants in non-familial APEs that are frequently encountered in clinical practice.

Targeted gene panels have been most frequently used for genetic testing as they are rapid and cost-efficient (*Lemke et al., 2012*). However, target genes must be limited to known mutations at the time of diagnosis, thus posing a challenging task with regard to keeping pace with newly identified genes after genetic testing, which often results in false-negative findings. Advancements in sequencing technology continuously and rapidly extend the list of novel epilepsy-causing genes and the cost of sequencing technology continues to drop. Therefore, WES or even whole genome sequencing offers substantial advantages in identifying potential causal epilepsy-related variants, particularly those of genetically undetermined non-familial epilepsies with heterogeneous phenotypes because new hypotheses for identifying novel epilepsy genes can be simply tested by reanalyzing previous WES or WGS data in silico.

### Genotype-phenotype correlation

The mTOR genes including *DEPDC5, TSC1* and *TSC2* have been associated with focal epilepsy, as was the case in our study in which the mTOR genes had the highest yield (13/32), although the yield was relatively low in some previous studies (*Lindy et al., 2018*; *Perucca et al., 2017*; *Carvill et al., 2013*). It is known that activating the mTOR pathway causes the epileptogenicity of brain malformations, specifically FCD, TS, and HS (*Liu et al., 2014*), which is supported by our finding that 80% of APEs with *TSC1* or *TSC2* variants had such malformations.

The ion channel-related genes are well-known to be the common genetic causes of early-onset epilepsies such as early-onset DEEs or genetic focal or generalized epilepsies, and were frequently identified as the presumed causative genes of epilepsy in most of the corresponding pediatric studies (*Lemke et al., 2012*; *Carvill et al., 2013*; *Kodera et al., 2013*; *Wang et al., 2014*; *Della Mina et al., 2015*; *Mercimek-Mahmutoglu et al., 2015*; *Hildebrand et al., 2016*; *Møller et al., 2016*; *Perucca et al., 2017*; *Dunn et al., 2018*; *Ko et al., 2018*; *Rim et al., 2018*; *Lee, Lee & Lee, 2018*). Ion channel-related genes had a higher yield (12/32) even in the present study, in which almost half of cases were adulthood-onset epilepsies. Given that five of 12 APEs with P/LPs of ion channel-related genes had adulthood-onset epilepsies, it seems plausible that these genes are implicated more frequently than expected in non-familial focal epilepsies in adulthood. While this needs to be functionally validated, it may widen our concept of the genetic spectrum of epilepsy in adulthood, which may in turn guide the development of adequate treatment options.

*ADGRV1* haploinsufficiency may be an important contributor to the development of genetic epilepsies, particularly those with myoclonic seizures (*Myers et al., 2018*). In our study, three APEs with the *ADGRV1* heterozygous variant (p.His1859Arg) had either focal or generalized epilepsy, which might be plausible if a focal myoclonic seizure was confused with a focal motor seizure, as is occasionally the case in outpatient clinics. *CNTNAP2* has been associated with cortical dysplasia-focal epilepsy syndrome (CDFES; OMIN#610042) or autosomal dominant epilepsy with auditory features (*Pippucci et al., 2015*). Although the original CDFES is an autosomal recessive trait, the APE (SU023 in Table 4) with the heterozygous *CNTNAP2* p.Ile172Thr variant in our study exhibited the typical CDFES features of FCD and focal epilepsy. The other APE with *CNTNAP2* p.Arg1288Cys had non-lesional focal epilepsy without auditory aura. A compound heterozygosity test using CNVkit was negative in these two cases. It is known that *EFHC1* Arg294His is a genetic cause of childhood absence epilepsy and juvenile myoclonus epilepsy (*Von Podewils et al., 2015*). However, APEs with *EFHC1* Arg294Cys, an allelic variant of Arg294His, had acquired posttraumatic epilepsy in our study. De novo heterozygous *PRICKLE1* variants have been linked to congenital brain malformations or myoclonic epilepsies (*Bassuk & Sherr, 2015*; *Todd & Bassuk, 2018*), while two of three APEs with *PRICKLE1* p.Ala541Ser variants in our study had non-lesional focal epilepsy and the other had acquired postencephalitic epilepsy. Although *RELN* has been associated with brain malformations and autosomal dominant lateral temporal lobe epilepsy, one of two APEs with *RELN* p.Thr1904Met variants had hippocampal sclerosis, one of the main pathological feature of mesial temporal lobe epilepsy, and the other had typical dermatological and radiological features of TS but the genetic test for *TSC1* or *TSC2* was negative. Further study is needed to elucidate whether *RELN* contributes to TS.

## Disease-modifying potential

The higher yield of genetic testing for familial epilepsies or early-onset DEEs has been associated with an earlier seizure onset or severity of the epilepsy (*Møller et al., 2016*; *Perucca et al., 2017*). However, such correlations were not evident in our study. This inconsistency may highlight characteristics of the genetic contribution to non-familial

epilepsies with a later age of onset or that are not so severe as to last until adulthood. Although most ion channel-related genes or mTOR genes have been associated with early-onset epilepsy syndromes with severe phenotypes such as Dravet's syndrome or intractable epilepsy with brain malformations that can lead to a grave outcome in early life, most APEs with P/LPs of ion channel-related genes or mTOR genes in our study experienced a later age of epilepsy onset or epilepsies that continued into adulthood. This suggests that these genes must somehow be linked to a disease-modifying mechanism that regulates the onset time or severity of the relevant epilepsy.

It is known that mTOR inhibitors such as rapamycin or everolimus have anti-epileptogenic effects rather than a simple seizure-suppression effect, as well as anti-tumor effects in TS (*Franz & Krueger, 2018*). Interestingly, in our study, all APEs with two P/LPs simultaneously had an mTOR gene variant. Although it requires validation in future studies, this finding, together with the mTOR inhibitors' modulating effects on epileptogenesis and tumor growth in TS (*Franz & Krueger, 2018*), suggests that mTOR genes are implicated in epileptogenesis or brain malformations (or both) as a key modulator of epistasis (gene-to-gene interaction). This could be supported by a recent report that *DEPDC5*, as a single mTOR gene, is a key contributor to a broad spectrum of lesional and non-lesional epilepsies, with variable but highly consistent phenotypes (*Baldassari et al., 2019*). Furthermore, considering that most APEs with *TSC1* or *TSC2* variants in our study experienced brain malformations and multi-drug resistant epilepsy for approximately 30 years on average, the notion of mTOR inhibitors with disease-modifying effects is a reminder of the importance of early identification of mTOR gene variants in patients with epilepsy or other dermatological mimics of TS to treat or halt disease progression.

Many of the P/LPs identified in our study were associated with atypical phenotypes or inheritance patterns that have not yet been reported in relation to their relevant epilepsies or epilepsy syndromes. This raises the possibility that the genetic basis of non-familial epilepsies, regardless of seizure onset time, differs from that of known familial epilepsies or pediatric DEEs. Given that five APEs with a mean seizure onset age of 44.2 years (range: 32–55 years) in whom possible genetic causes were identified had definitely acquired etiologies prior to seizure onset, including traumatic brain tissue loss or encephalitis, it is plausible that one variant of the relevant genes (*SCN1A, TSC1, ADGRV1, EFHC1* and *PRICKLE1*) alone may not be sufficient to cause the relevant epilepsies in the absence of acquired brain damage. This also reinforces the implication of disease-modifying factors— whether they are genetic, environmental, or something yet to be identified—in the pathogenesis of epilepsies that start in or last until adulthood.

## Pathogenic potential of EAGs in epilepsy pharmacoresistance

It is known that *SCN1A* variants are associated with poor surgical outcomes and CBZ-induced seizure aggravation (*Franco & Perucca, 2015*; *Skjei et al., 2015*). In our study, the treatment response to CBZ varied according to individual variants, suggesting that *SCN1A*-associated drug responsiveness may be an allele-specific phenomenon, not gene-specific, although this is inconclusive due to the small sample size. Nevertheless, the

results of our study can be used to guide a trial to halt CBZ use in APEs with multi-drug resistance.

Unlike our expectation, the target hypothesis of epilepsy pharmacoresistance was not verified in our study. Instead, most APEs with P/LPs of neurodevelopment-associated epilepsy genes such as *TSC2* or *RELN*, or with structural brain lesions, were multi-drug resistant. This suggests that pharmacoresistance in APEs may, at least in part, be linked to neural network rearrangement by structural lesions or potential somatic mutations in situ. An international collaboration of epilepsy studies could uncover these results.

The present study has several limitations. First, WES is not the best option for detecting copy number variants, large-sized indels, trinucleotide repeats, intronic alterations, intergenic variants, structural chromosomal rearrangements, or epigenetic modifications associated with epilepsy (*Biesecker & Green, 2014*). This suggests that the diagnostic yield of our study may be the minimum yield possible with WES for non-familial APEs. Second, although all identified P/LPs are seemingly post-zygotic de novo mutations defined by the absence of family history of epilepsy, the possibilities of unknown family histories, somatic mutation, genetic mosaicism, or lower penetrance were not validated owing to limitations in DNA or tissue sampling. Third, although the variants were selected via a customized stringent filtering process and classified as pathogenic or likely pathogenic according to ACMG guidelines, the pathogenicity of each variant should be confirmed in future studies. Fourth, this study selected target genes for analysis from known epilepsy-related genes, which precludes the chance to identify novel epilepsy genes. However, detecting mutations in known epilepsy genes in patients with an uncommon or unspecific presentation of a seizure disorder may help reveal the true phenotypic spectrum of the disorder (*Lemke et al., 2012*).

## CONCLUSIONS

Our study possibly reveals causal genetic variants in 13.2% of non-familial patients with predominantly focal epilepsy in which mTOR genes and ion channel-related genes are most commonly associated. These potentially pathogenic variants, identified in the genes that have been associated with early-onset epilepsies with severe phenotypes, were also linked to epilepsies that start in or last until adulthood in this study, thereby suggesting the implication of one or more disease-modifying factors that regulate the onset time or severity of the disease during epileptogenesis. Neurodevelopment-associated epilepsy genes, such as *TSC2* or *RELN*, or structural brain lesions were more strongly associated with epilepsy pharmacoresistance. Our results highlight the importance of earlier identification of the genetic etiology of non-familial epilepsies in adulthood, leading us to the best treatment option in terms of precision medicine and to future neurobiological research for novel drug development.

## ACKNOWLEDGEMENTS

We are grateful to the patients for their help and participation in the study. We thank Hee-Joo Kim and Sun-Ok Lee for technical assistance. We would like to thank Editage for English language editing.

### Funding

This research was supported by a grant of the Korea Health Technology R&D Project through the Kores Health Industry Development Institute (KHIDI), funded by the Ministry of Health & Welfare, Republic of Korea (Grant Number: HI15C1559). The funders had no role in study design, data collection and analysis, decision to publish, or preparation of the manuscript.

### Grant Disclosures

The following grant information was disclosed by the authors:
Korea Health Technology R&D Project through the Kores Health Industry Development Institute (KHIDI).
Ministry of Health & Welfare, Republic of Korea: HI15C1559.

### Competing Interests

The authors declare that they have no competing interests.

### Author Contributions

- Kyung Wook Kang performed the experiments, analyzed the data, contributed reagents/materials/analysis tools, prepared figures and/or tables, authored or reviewed drafts of the paper, approved the final draft, phenotyping.
- Wonkuk Kim conceived and designed the experiments, analyzed the data, contributed reagents/materials/analysis tools, authored or reviewed drafts of the paper, approved the final draft.
- Yong Won Cho performed the experiments, approved the final draft, phenotyping.
- Sang Kun Lee performed the experiments, approved the final draft, phenotyping.
- Ki-Young Jung performed the experiments, approved the final draft, phenotyping.
- Wonchul Shin performed the experiments, approved the final draft, phenotyping.
- Dong Wook Kim performed the experiments, approved the final draft, phenotyping.
- Won-Joo Kim performed the experiments, approved the final draft, phenotyping.
- Hyang Woon Lee performed the experiments, approved the final draft, phenotyping.
- Woojun Kim performed the experiments, approved the final draft, phenotyping.
- Keuntae Kim performed the experiments, approved the final draft, phenotyping.
- So-Hyun Lee performed the experiments, contributed reagents/materials/analysis tools, prepared figures and/or tables, approved the final draft.
- Seok-Yong Choi conceived and designed the experiments, analyzed the data, contributed reagents/materials/analysis tools, prepared figures and/or tables, authored or reviewed drafts of the paper, approved the final draft.
- Myeong-Kyu Kim conceived and designed the experiments, performed the experiments, analyzed the data, contributed reagents/materials/analysis tools, prepared figures and/or tables, authored or reviewed drafts of the paper, approved the final draft.

## Human Ethics

The following information was supplied relating to ethical approvals (i.e., approving body and any reference numbers):

This study was approved by the institutional review boards at Chonnam National University Hospital (CNUH-2016-028).

## Data Availability

Data is registered at CODA (Clinical-Omics Data Archive; http://coda.nih.go.kr). Registration No.: R000051, R000374, R000854, R001354.

CODA data is also available at figshare: Kang, Kyung-Wook; Kim, Wonkuk; Yong Cho, Won; Kun Lee, Sang; Jung, Ki-Young; Chul Shin, Won; et al. (2019): Genetic characteristics of non-familial epilepsy. figshare. DOI 10.6084/m9.figshare.9988172.

## Supplemental Information

Supplemental information for this article can be found online at http://dx.doi.org/10.7717/peerj.8278#supplemental-information.

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
