# Peer review of "Genetic characteristics of non-familial epilepsy"

_PeerJ, doi:10.7717/peerj.8278_

## Round 0.1 · original submission · Minor Revisions

· Academic Editor

Minor Revisions

Dear Authors, Please revise your manuscript as per suggestions from the three peer reviewers as soon as possible.

Reviewer 1 ·

Basic reporting

This is a very ambitious, technically solid, manuscript that addresses the relevant question of the role of genes in non-familial cases of epilepsy. It has the potential to be a high impact paper. However, improvements are required to reach the target of readability that deserves. The first improvement is certainly in the language. The article must be written in better English and must use clear, unambiguous, technically correct text to conform to professional standards of courtesy and expression. I suggest that you find a native English language colleague to review your manuscript. Expressions like “Contrary to neonatal to childhood epilepsy”, “There might be hidden genetics in non-familial APEs”, and many others can be improved. The use of undefined acronyms, like mTORGs for “mammalian target of rapamycin (mTOR) pathway-related genes” or ICRGs for “ion channel-related genes” makes the manuscript less readable. The article includes a sufficient introduction and background to demonstrate how the work fits into the broader field of knowledge. Relevant prior literature is also appropriately referenced. The structure of the article also conforms to an acceptable format. Figures are relevant to the content of the article and appropriately described and labeled. All appropriate raw data have been made available.

Experimental design

The research question is well defined, relevant, meaningful, and within the aims and scope of the journal. The authors also make clear how their research contributes to filling an identified knowledge gap in the field. The investigation was also conducted rigorously and with a high technical standard, in conformity with the prevailing ethical standards in the field. Methods are described with sufficient information to be reproducible by other investigators.

Validity of the findings

Impact and novelty of the manuscript are potentially very high, however, the authors should discuss better their results and point to the novelties they have found. All underlying data are provided; they are robust, statistically sound, and controlled. The data on which the conclusions are based are solid. Conclusions are well stated but a little confusing, linked to the original research question and limited to those supported by results. Claims of a causative relationship are supported by clear evidence.

Additional comments

The manuscript summarizes an impressive amount of work that deserves a better presentation. Improvements in the language and the fluency of the manuscript require some additional efforts. This will certainly benefit the readability of the manuscript and will make it more appreciated by other investigators in the field.

·

Basic reporting

no comment

Experimental design

No comment

Validity of the findings

no comment

Additional comments

This is an interesting non-familial epilepsy study by using whole exome sequencing to study the genetic characteristics in a total of 243 patients. The study hypothesis and method were well written and relative good description in the results and discussion sessions. Some points need to be raised:

1. It is pity that familial segregation was not performed in those 23 non-familial patients with presumed disease-causing variants. The mutation status in other family members can provide the information concerning whether the mutation was de novo or the gene penetration maybe incomplete.

2. In patient SU023 having heterozygous CNTNAP2 p.Ile172Thr variant who presented with typical CDFES, I wonder if there is information showing large deletion or copy number variations from the WES data? Certain analyses (such as cn.mops (Copy Number estimation by a Mixture Of PoissonS)) can be performed to retrieve those information. Likewise, similar analysis should be considered in other individuals.

3. I would like to know, for patients with TSC1/TSC2 variants, do they fit the diagnostic criteria of tuber sclerosis complex? What is the clinical presentation in other organ, including the skin, heart, kidneys, and lung?

4. As stated, the identified pathogenic variant numbers were too small for further analysis in the drug sensitive and resistance, which was the original hypothesis of this manuscript. This indicated that the goal cannot be achieved without international collaboration of epilepsy studies.

Reviewer 3 ·

Basic reporting

no comment

Experimental design

no comment

Validity of the findings

no comment

Additional comments

Through a large cohort of adult patients with non-familiar epilepsy, aim of the authors was to better understand genetic basis of this huge amount of epileptic patients without a definite aetiology. The paper is original and of great interest because in the last years most of articles have been published on DEEs, while genetic basis of adult and non-familiar cases remained unclear. Overall the paper is original, well written, fluent and clear. However, there are few comments consider before publication:

1. The definition of DR (drug-resistant) and DS (Drug-responsive) is not clear. There is the group of patients with less than 12 seizures/year that does not fit with both groups. Is this definition arbitrary or did you refer to literature? Please clarify.
2. Half of the study population is DR. This rate is higher (usually DR patients are 30%). Please comment this result in the discussion giving possible hypothesis for this result (age, aetiology…).
3. Ages at onset and at study should be of more precise also in the text (only in table 3). Please clarify the sentence “ages at recruitment and at seizure onset were approximately 40 and 20 years” including medians and ranges for each age.
4. Authors refer to “Epileptic Encephalophaties (EEs)”. It would be better to us the updated definition of “Developmental and Epileptic Encephalophaties (DEEs).

---

## Round 0.2 · accepted · Accept

· Academic Editor

Accept

Dear Authors,Good news! Your revised manuscript is currently suitable to be published in PeerJ.

Reviewer 1 ·

Basic reporting

The paper has been profoundly improved in the language that is now unambiguous and technically correct. It conforms to professional standards and is certainly more easily readable.
References, structure, figures, tables, and raw data are in excellent shape.
The results are now clear and better appreciated.

Experimental design

It is sharp and clearly defined. The authors have conducted their research rigorously with excellent technical standards. Language improvement has undoubtedly helped to understand better the experimental design and the superb work done.

Validity of the findings

The findings are relevant and original. Conclusions are well stated.

Additional comments

The paper is now in good shape and it is suitable for publication. The authors have done a good job. Their excellent work deserved an effort to improve the presentation.

·

Basic reporting

no comment

Experimental design

no comment

Validity of the findings

no comment

Additional comments

The authors responded to my questions point by point. I am satisfied with authors' reply.